# Role of the αC-β4 loop in protein kinase structure and dynamics

Jian Wu[1†], Nisha A Jonniya[1†], Sophia P Hirakis[2], Cristina Olivieri[3], Gianluigi Veglia[3,4], Alexandr P Kornev[1], Susan S Taylor[1,2]*

[1]Department of Pharmacology, University of California, San Diego, San Diego, United States; [2]Department of Chemistry and Biochemistry, University of California, San Diego, San Diego, United States; [3]Department of Biochemistry, Molecular Biology, and Biophysics, University of Minnesota, Minneapolis, United States; [4]Department of Chemistry, University of Minnesota, Minneapolis, United States

## eLife assessment

This **valuable** study draws attention to the importance of a previously overlooked structural motif in kinase regulation. While the data presented are intriguing and mostly **solid**, further analysis and additional experiments will be needed in the future to support the authors' hypothesis. The work will be of interest to protein biochemists and enzymologists with an interest in kinases and allostery.

*For correspondence:
staylor@ucsd.edu

†These authors contributed equally to this work

Competing interest: The authors declare that no competing interests exist.

**Abstract** Although the αC-β4 loop is a stable feature of all protein kinases, the importance of this motif as a conserved element of secondary structure, as well as its links to the hydrophobic architecture of the kinase core, has been underappreciated. We first review the motif and then describe how it is linked to the hydrophobic spine architecture of the kinase core, which we first discovered using a computational tool, local spatial Pattern (LSP) alignment. Based on NMR predictions that a mutation in this motif abolishes the synergistic high-affinity binding of ATP and a pseudo substrate inhibitor, we used LSP to interrogate the F100A mutant. This comparison highlights the importance of the αC-β4 loop and key residues at the interface between the N- and C-lobes. In addition, we delved more deeply into the structure of the apo C-subunit, which lacks ATP. While apo C-subunit showed no significant changes in backbone dynamics of the αC-β4 loop, we found significant differences in the side chain dynamics of K105. The LSP analysis suggests disruption of communication between the N- and C-lobes in the F100A mutant, which would be consistent with the structural changes predicted by the NMR spectroscopy.

## Introduction

Although the protein kinases, like the GTPases, have evolved to be highly regulated molecular switches, they transfer the γ-phosphate of adenosine triphosphate (ATP) to a protein substrate instead of to water. Understanding how mature and fully active protein kinases couple ATP binding to peptide/protein binding is especially challenging as it involves many sites that lie distal to the active site. This process of phosphorylating a heterologous protein substrate should be clearly distinguished from *cis*-autophosphorylation of the kinase core, which is usually a key initial step in the assembly of most active kinases. *Cis*-autophosphorylation of a protein kinase core is distinct and different from transferring the phosphate to a heterologous protein. In cells, a fully active kinase transfers the γ-phosphate of ATP to a heterologous protein substrate that is typically tethered to a distal site that lies far from the site of phosphoryl transfer. To ask how a mature and fully active protein kinase couples ATP binding to peptide/protein binding we use the catalytic (C) subunit of cAMP-dependent protein kinase (PKA)

as a model system. PKA activity is regulated by inhibitory regulatory (R) subunits and by heat-stable protein kinase inhibitors (PKIs). In addition, we have extensive NMR data and computational analyses of PKA that complement crystal structures and provide essential windows into dynamics.

The conserved motifs that define the kinase core were first recognized when Dayhoff aligned the cloned sequence of Src with the manually sequenced C-subunit of PKA in 1982 (*Barker and Dayhoff, 1982*). By manually aligning a handful of protein kinases, Hanks, et al, subsequently showed that these motifs, which were scattered throughout the kinase core, were conserved in all protein kinases (*Hanks et al., 1988*). When the first protein kinase structure was solved in 1991, these conserved sequence motifs became structural entities that correlated with β strands, α helices, and loops of the folded protein (*Knighton et al., 1991a*). That first structure, which was actually a binary complex of the PKA C-subunit and an inhibitory peptide (IP20) from PKI, also provided a detailed description of how the IP20 peptide was docked onto the kinase core (*Knighton et al., 1991b*). Subsequent structures of a ternary complex with ATP and IP20, solved in 1993, showed how the conserved motifs correlated with ATP and peptide binding and engaged the entire kinase core (*Zheng et al., 1993*; *Bossemeyer et al., 1993*). Although Johnson provided a more detailed description of these motifs (*Johnson et al., 2001*), several essential points were not yet understood. Only later, using computational tools, did we come to appreciate how the conserved and non-linear hydrophobic core architecture was assembled and how hydrophobicity correlated with allosteric regulation of the kinase core (*Kornev et al., 2006*; *Kornev et al., 2008*; *Kim et al., 2017*). In addition, we did not initially appreciate the importance of the αC-β4 loop. Like helices and strands, the αC-β4 loop is also a well-defined element of secondary structure (*Rose, 2023*; *Rose et al., 1983*), and this loop in the N-terminal lobe (N-Lobe) of the kinase core is essential for mediating the synergistic high-affinity binding of ATP and IP20.

Unlike other protein kinases such as PKC and the Leucine-rich repeat kinase 2 (LRRK2) that are flanked by other domains that regulate activation and subcellular localization (*Newton, 2018*; *Taylor et al., 2022*; *Weng et al., 2023*), the PKA C-subunit represents a kinase domain that is flanked by relatively short N- and C-terminal tails (*Kannan et al., 2007*). In cell the C-subunit is then assembled as an inactive holoenzyme with functionally non-redundant cAMP binding regulatory (RIα, RIβ, RIIα, and RIIβ) subunits allowing the activity of the kinase domain to be unleashed rapidly and reversibly by allosteric binding of cAMP to the cyclic-nucleotide binding domains (CNB) of the R-subunits (*Taylor et al., 2012*). In addition, the PKA C-subunit can be regulated by PKI. The R-subunits and PKI all share an inhibitor site that docks to the active site cleft of the C-subunit; however, PKI and RIα/RIβ are unusual in that they are pseudo substrates where high-affinity binding is synergistically coupled to the high-affinity binding of ATP (~60 nM). Even though the structure of the PKA C-subunit was solved over 30 years ago, we still are elucidating the mechanistic details of how the highly dynamic features of the kinase core are regulated by a set of well-defined motifs and, in particular, how the high-affinity binding of ATP and IP20 are coupled. Disease mutations are also providing key mechanistic insights into how the synergistic binding of ATP and PKI can be uncoupled. Using the PKA C-subunit as a prototype, we focus here, in particular, on the conserved hydrophobic residues/motifs that define the core protein kinase architecture, which contributes so significantly to entropy-driven allostery (*Kornev and Taylor, 2015*). We then describe how the αC-β4 loops is anchored to the hydrophobic core architecture. Finally, we explore how a single mutation, F100A in the αC-β4 loop, uncouples the synergistic binding of ATP and PKI using a computational approach that first identified the hydrophobic spines (*Kornev et al., 2006*; *Kornev et al., 2008*).

## Discovering the hydrophobic core architecture of protein kinases

While we elucidated conserved motifs in the kinase core early on *Barker and Dayhoff, 1982*; *Hanks et al., 1988*; *Johnson et al., 2001* and identified, using affinity labeling, specific residues that we hypothesized were associated with ATP binding (*Zoller and Taylor, 1979*; *Buechler and Taylor, 1989*), we did not have a thorough understanding of the bilobal kinase fold until the first protein kinase structure was solved. This first structure defined the N-Lobe and the C-Lobe and validated the active site localization of the regulatory triad that was identified by affinity labeling and cross-linking. This triad consists of two residues in the N-Lobe - K72 in β3 and E91 in the αC-Helix and one in the DFG motif in the C-Lobe - D184 (*Knighton et al., 1991a*). The subsequent structures in 1993 (*Zheng et al., 1993*; *Bossemeyer et al., 1993*) showed how ATP was held into the active site cleft between the two lobes. Together, these structures validated the affinity labeling experiments where the three motifs

containing the regulatory triad residues converged on the ATP phosphates and $Mg^{2+}$ ions at the active site cleft. This structure of a fully closed conformation also defined a novel ATP binding motif where the adenine ring of ATP was buried under a Glycine-rich Loop (G-loop) in a pocket at the base of the cleft (*Zheng et al., 1993*; *Bossemeyer et al., 1993*). From these early structures, we gain an appreciation for what is now called the 'Activation Loop.' The Activation loop is typically assembled by a key phosphorylation site (*Nolen et al., 2004*; *Johnson et al., 1996*), and this loop is very stable in the PKA C-subunit because T197 is constitutively phosphorylated in the purified C-subunit. At that time, however, we did not appreciate the highly dynamic features that regulate kinase activity, including the Activation loop, nor did we appreciate the role of hydrophobicity in driving dynamics. This would require both computational tools that were not yet sufficiently robust as well as nuclear magnetic resonance (NMR) spectroscopy. In addition, we would need structures of other kinases where the Activation loop is typically disordered (*Nolen et al., 2004*; *Johnson et al., 1996*).

The importance of hydrophobic residues and the concept of non-contiguous but spatially conserved hydrophobic motifs, some highly conserved and others just conserved as hydrophobic residues, did not become obvious until we investigated structural differences with a computational approach called LSP alignment. With LSP alignment we first identified spatially conserved residues that revealed a conserved hydrophobic 'spine' architecture that was associated with active kinases but broken into inactive kinases (*Kornev et al., 2006*). The terminology of 'spines' is related to the fluidity of these residues in contrast to hydrophilic residues and ion pairs that are locked into a more rigid conformation by hydrogen bonds or electrostatic bonds. The Regulatory Spine (R-spine) residues were identified first and correspond to four spatially conserved residues - two in the C-lobe and two in the N-lobe that were aligned in every active kinase but broken in inactive kinases (*Kornev et al., 2006*). The assembly of the R-Spine defines the switch mechanism of every active kinase. It reveals how kinases have evolved to be dynamic molecular switches, similar to the GTPases. In contrast to metabolic enzymes, they have not evolved to be efficient catalysts. The highly regulated R-Spine along with the Regulatory Triad (K72/E91/D184) remain as hallmark

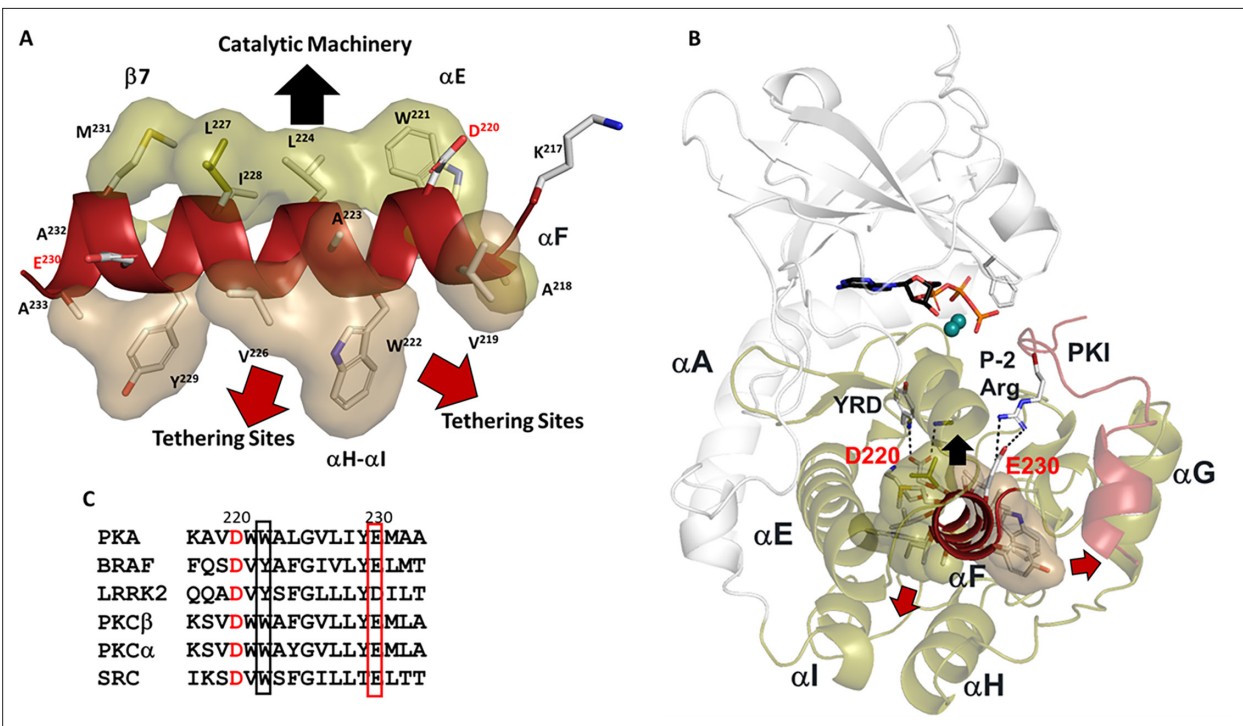

**Figure 1.** Hydrophobic αF-helix serves as a central scaffold. (**A**). αF-helix is very hydrophobic and creates an interface with multiple motifs including the Catalytic machinery (colored in tan) and tethering sites (in sand) of protein kinase (PKA) C-subunit. Two key charged residues, D220 and E230, sit at the two ends of αF-helix. (**B**). αF-helix is a central scaffold for the assembly of the entire molecule. D220 forms two H-bonds to Y164 and R165 of YRD motif, and E230 salt-bridges to P-2 Arg of PKI. (**C**). Sequence alignment of αF-helix segment of PKA with other kinases. All share a very hydrophobic helix, the highly conserved residues, D220 are colored in red and E230 in red box.

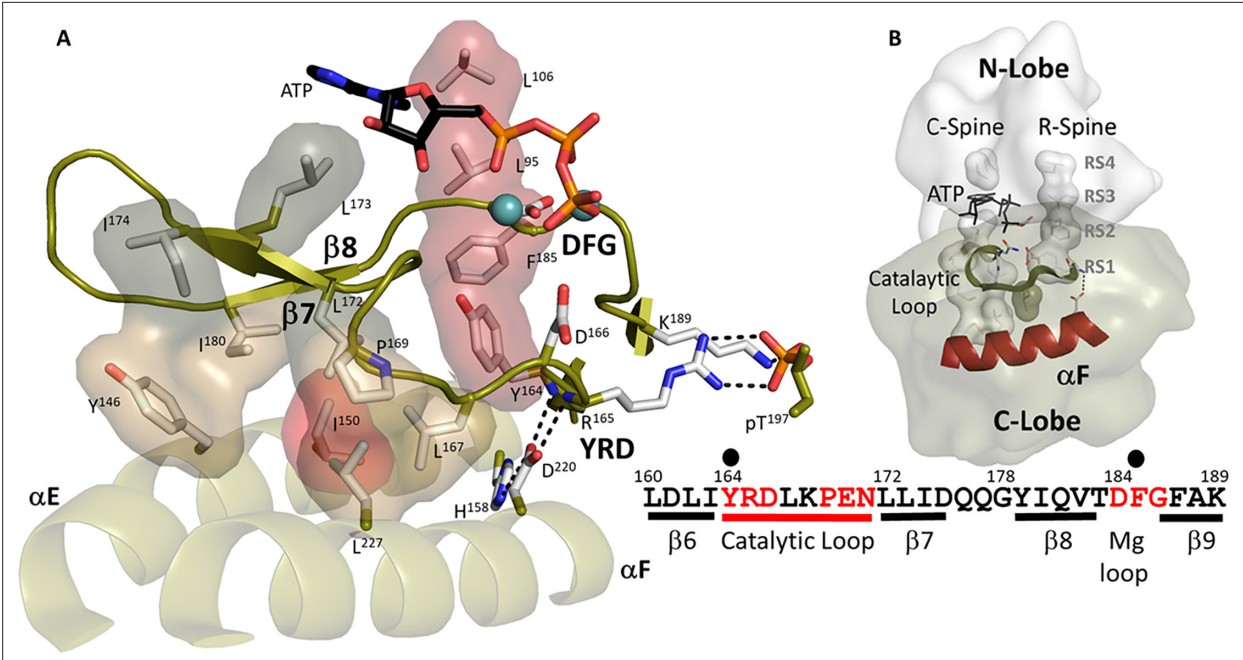

**Figure 2.** Hydrophobic interface anchors the Catalytic machinery to the αE-helix and αF-helix. (**A**). I150 (in red) from αE-helix plays an important role by docking to αF-helix and Catalytic loop. Three residues from β7, L172, L173, and I174 (in dark tan) assemble the hydrophobic surface from αE-helix to adenosine triphosphate (ATP) pocket. L173 and I174 are part of C-Spine. R-spine is also shown in red. D220 bridges H158 from αE-helix to YRD motif. (**B**). The logo of spines shows how important those hydrophobic residues are. The sequence of β6-β9 segment is also shown.

signature motifs of every active kinase that is capable of trans-phosphorylation of a heterologous protein substrate.

A second motif, referred to as the Catalytic Spine (C-Spine), defines the extensive and fundamental core hydrophobic architecture of every kinase domain (*Kornev et al., 2008*). The C-spine includes motifs in both the N-Lobe and the C-Lobe, which are connected by the hydrophobic capping of the adenine ring of ATP. Capping of the adenine ring is accomplished by two highly conserved residues (A70 in β3 and V57 in β2) in the N-Lobe while the other surface of the adenine ring is capped by a C-Lobe residue, L173 in β7, which is a conserved hydrophobic residue found in all kinases. The dominant feature of the C-spine is the very unusual hydrophobic αF-Helix that spans the C-lobe and is linked to all of the important elements of the C-Lobe (*Figure 1*). This buried helix is unusual for several reasons. With a conserved glycine in the middle, it does not have a strong helical propensity. In general, it is highly unusual to find a buried hydrophobic helix in a globular domain; in many ways, it is more like a trans membrane helix. The αF helix is flanked by two charged residues. At the beginning of the αF Helix is the highly conserved D220 that couples the αF Helix to the catalytic machinery at the active site cleft (*Figure 2*). Specifically, it hydrogen bonds to the backbone amides of Y164 and R165 in the Y/HRD motif that precedes the catalytic loop. D220 is followed by another highly conserved residue, W222, which faces the αH-αI loop, a tethering site on the bottom of the C-Lobe (*Deminoff et al., 2006*; *Deminoff et al., 2009*). W222 also shields the conserved ion pair between E208 at the end of the APE motif and R280 in the αH-αI loop. At the other end of the αF Helix is E230 which in PKA recognizes the P-2 arginine in the substrate peptide/protein. The remarkable features of the αF-Helix that allow it to be the central organizing unit of the C-lobe, summarized in *Figure 1*, were clearly revealed in the LSP plots in 2008 (*Kornev et al., 2008*). What was not fully appreciated in these first analyses, however, was the role of the αE-Helix in the C-Lobe and another highly conserved motif in the N-Lobe, the αC-β4 loop.

## The αC-β4 Loop

The two spine residues in the N-Lobe, which lie at the end of the αC Helix (RS3) and the beginning of β4 (RS4), stabilize a very important element of secondary structure that is conserved in every protein kinase - the αC-β4 loop (residues 99–106) (*Figure 3*). The two R-spine residues are brought together

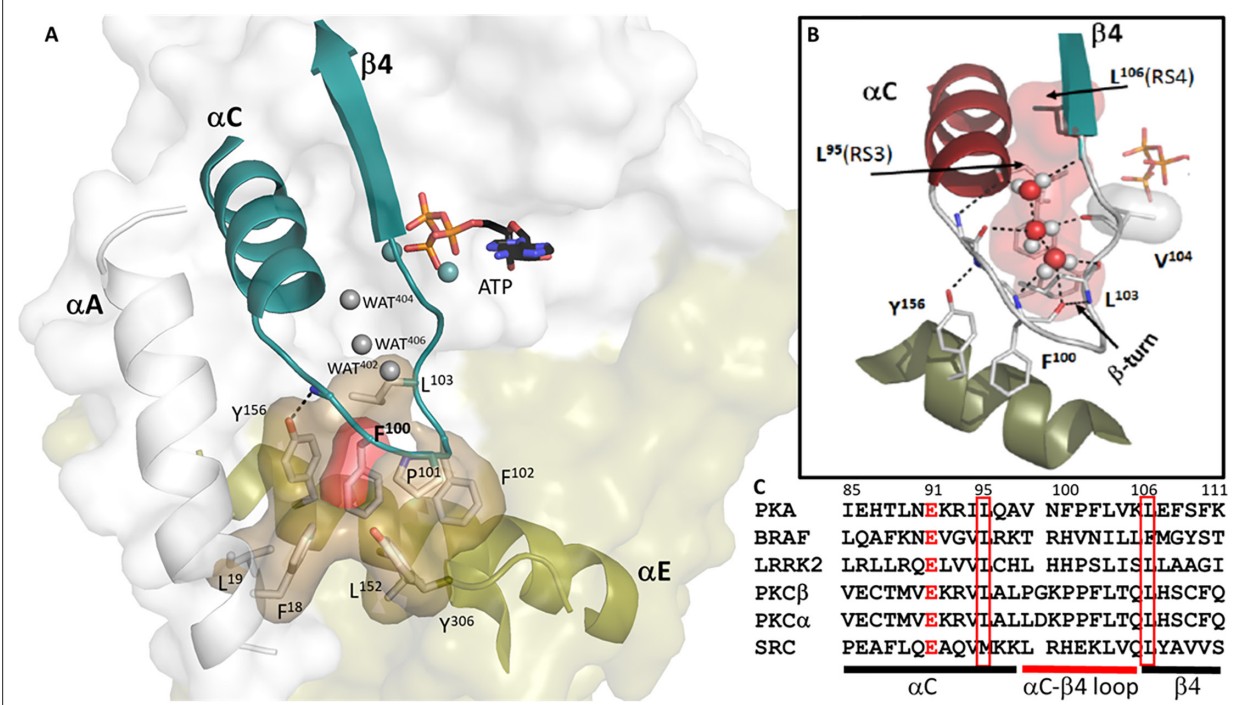

**Figure 3.** The buried surface of αC-β4 loop. (**A**). αC-β4 loop (in teal) in protein kinase (PKA) links the N-lobe to its C-lobe. The hydrophobic residues on the loop docks to αE-helix, F18, and L19 from αA-helix and Y306 from C-tail are also part of this surface. F100 is colored in red. (**B**). The H-bond network of αC-β4 loop. Y156 from αE-helix H-bonds to N99 backbone amide. Three water molecules also help to nucleus the network. (**C**). Sequence alignment of αC-β4 loop of PKA with other kinases. The highly conserved residues are highlighted, E91 colored in red, L95 and L106 in red box.

by a strategic-β-turn motif (residues 100–103) where the carbonyl of F100 is hydrogen bonded to the backbone amide of L103 (*Rose, 2023*). The other carbonyls and amides of this turn, as well as those of the other residues in this motif, are filled by ordered water molecules (*Figure 3B*). Structured water molecules highlight that this surface of the αC-β4 loop is exposed to solvent, while the other buried surface of the αC-β4 motif is very hydrophobic and anchored to the hydrophobic R-Spine and Shell residues (*Meharena et al., 2013*). Only one residue in the αC-β4 motif, V104, directly touches ATP through one of its methyl side chains that is anchored to the adenine ring.

Although highly conserved and very stable, this β-turn is not traditionally recognized as a stable element of secondary structure (*Rose et al., 1983*). The two N-lobe spine residues (RS3 and RS4) anchor the αC-β4 motif to the R-Spine residues in the C-Lobe (*Figure 3B*) while the tip of the loop is very hydrophobic (*Figure 3A*). The critical features at the tip of the β-turn typically include a proline. Even though the backbone amides and carbonyls of the β-turn are solvent exposed, one can see in PKA that the hydrophobic side chain residues at the tip of the loop, F100-P101-F102-L103, are buried in a hydrophobic pocket comprised primarily of residues in the αE-Helix. The strategic importance of stabilizing the backbone of this αC-β4 loops is highlighted by Y156 in the αE helix, in which hydrogen bonds to the backbone amide of N99. Although this residue (N99) is not conserved, its functional importance has been highlighted by BRaf where replacement of this residue (R509) with histidine breaks the BRaf dimer interface that is essential for BRaf activation (*Hu et al., 2013*; *Shaw et al., 2014*). The importance of this residue was also explored in depth for the EGF receptor where this site is a hot spot for oncogenic mutations (*McSkimming et al., 2015*; *Ruan and Kannan, 2018*). Although the residue at the N99 position differs in every kinase, its spatial organization as well as the hydrogen bonding of its backbone to the αE helix is conserved, and its strategic function may also be conserved.

The first 600 picosecond Molecular Dynamics (MD) simulations of the PKA C-subunit, published in 1999, not only demonstrated that the N-Lobe and the C-Lobe function as independent rigid bodies but also showed that the αC-β4 loop (residues 99–106) is the only piece of the N-lobe that remains anchored to the C-lobe whether the kinase is in an open or closed conformation (*Tsigelny et al., 1999*; *Figure 4*). We now know from many kinase structures that positioning and organizing the N-lobe for

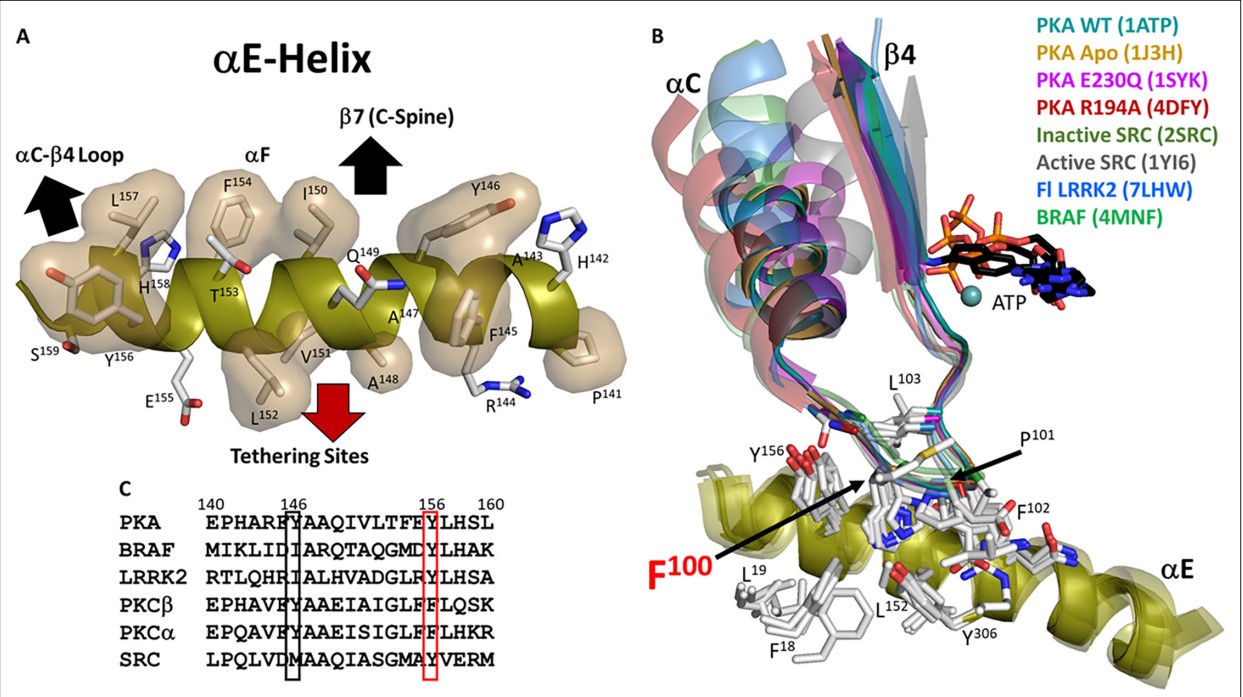

**Figure 4.** αC-β4 loop is a very stable element. (**A**). Hydrophobic surface of αE-helix which anchors to αC-β4 loop, αF-helix, and tethering sites. (**B**). Superimposition of αC-β4 loop in structures of protein kinase (PKA) and other kinases. αC-β4 loop is very stable, not only in different conformations of PKA, but also in other kinase structures including active and inactive Src. (**C**). Sequence alignment of αE-helix of PKA with other kinases. Y156, in red box, is highly conserved.

catalysis is complex and highly regulated. The analysis of LRRK2 supports the prediction that the N- and C-lobes of the kinase core function as independent rigid bodies (*Myasnikov et al., 2021*; *Zhu et al., 2022*). It also indicates that the αC-β4 loop remains as one of the most stable elements of the kinase core whether LRRK2 is in an active or an inactive conformation (*Figure 4*). The αC-β4 loop is also stably anchored to the αE helix in the active and inactive Src as well as in the active and inactive BRAF (*Figure 4*). Indeed, the αC-β4 loops is conserved as a stable element of secondary structure in every protein kinase as it anchors to the αE Helix.

## The αE Helix

Although the dynamic αC Helix, as well as the stable αF Helix described above (*Figure 1*), have been long recognized as conserved elements of the kinase core, less attention has been given to the αE Helix. A defining feature of the αE Helix, mentioned above, is Y156. This conserved tyrosine, which can also be a histidine or phenylalanine in other kinases, is anchored to the backbone amide of N99 in the αC-β4 loop. The importance of this residue is highlighted in BRAF, as discussed previously. Two residues down from Y156 is H158 which in PKA is firmly anchored to the conserved D220 in the αF Helix. H158 is one of three dually protonated histidines in the C-subunit (*Gerlits et al., 2019*; *Wych et al., 2023*). In the fully active C-subunit that is phosphorylated on its Activation loop the side chain of His158 anchors the backbone amides of the HRD residues (R165 and D166) in the Catalytic loop. In contrast to the Activation loop (residues 184-pT197), which is dynamically assembled, the backbone of the catalytic machinery in the C-Lobe (residues 162–182) is very stable. As seen in *Figure 2*, this interaction as well as the hydrophobic anchoring of the catalytic loop to the αE and αF Helices makes the backbone of the catalytic loop very stable. H158 is conserved as a histidine or phenylalanine in most kinases and in the EGF receptor and in other tyrosine kinases this region is a hot spot for oncogenic mutations (*Weng et al., 2023*).

I150 in the αE Helix is also a key residue. The importance of I150 as a critical player in the hydrophobic core architecture was first highlighted by NMR (*Kim et al., 2017*). As seen in *Figure 2*, I150 is important for many reasons. It is anchored to L167 and P169 in the Catalytic loop and to L172 in

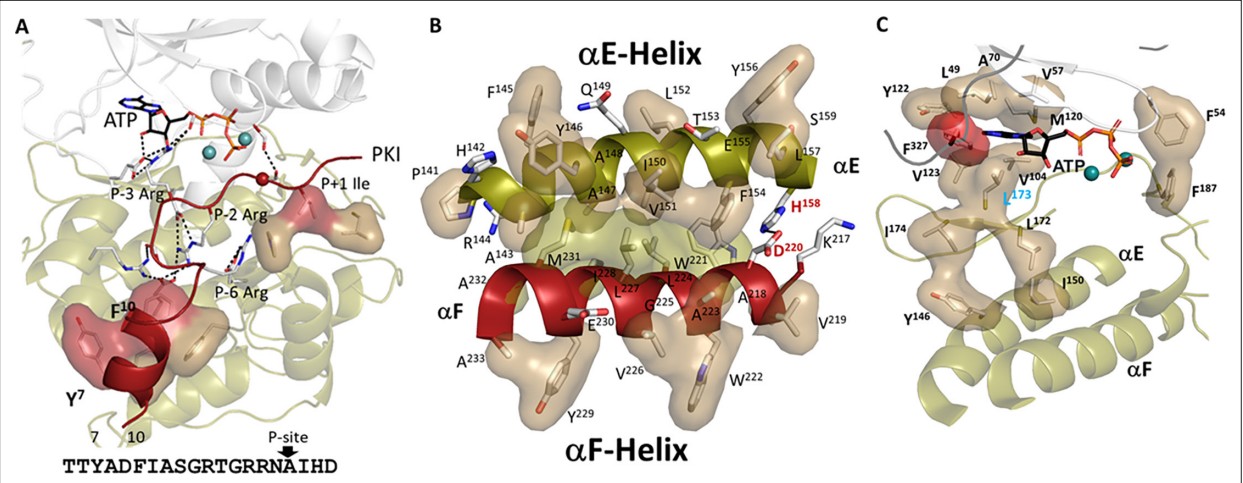

**Figure 5.** Hydrophobic residues play key roles in protein kinase (PKA). (**A**). High-affinity binding of protein kinase inhibitor (PKI) is mediated by the hydrophobic surface of an amphipathic helix and *P*+1 inhibitor site, both highlighted in red. The sequence of PKI is also shown. (**B**). Hydrophobic interface between the αE and αF-helices. One side of αF-helix (in red) is shown in tan, and another side in sand, which is the same color coding as *Figure 1A*. (**C**). Hydrophobic pocket surrounding ATP. L172, L173, and I174 from β7 anchor this adenosine triphosphate (ATP) pocket to αE-helix. F327 from C-tail is highlighted in red.

β7, which connects the adenine capping residue, L173, to the hydrophobic core. On the other side of L173, both I174 and I180 in β8 anchors the adenine capping residue to the αE Helix. In contrast to the Activation Segment and the R-spine, which are dynamically assembled as part of the activation, the backbone residues of the catalytic machinery are always very stably anchored to the hydrophobic core architecture. The two R-Spine residues in the C-Lobe, RS1 in the HRD motif and RS2 in the DFG motif, are all highlighted in *Figure 2* where we can appreciate that the catalytic machinery is anchored to the R-Spine, similar to the hydrophobic surface of the αC-β4 loop.

## Synergistic high-affinity binding of ATP and pseudo-substrate inhibitors

A highly unusual allosteric feature of PKA that is perhaps unique to PKA is its heat-stable protein kinase inhibitor (PKI). PKI was first identified as a high-affinity inhibitor of the PKA C-subunit in 1971 shortly after the C-subunit was discovered (*Walsh et al., 1971*). It is a classic intrinsically disordered protein (IDP), a prediction that was fully validated recently by NMR of full-length PKI (*Olivieri et al., 2020*). Once it was sequenced (*Scott et al., 1985b*), PKI was shown to be a pseudosubstrate, a feature that is shared by the type I Regulatory (R) Subunits of PKA, RIα and RIβ. While the inhibitory properties of PKI were localized to the N-terminus (residues 5–24) (*Scott et al., 1985a*; *Cheng et al., 1985*), the unusual synergistic high-affinity binding of ATP and PKI was characterized in detail by the peptide studies of *Whitehouse and Walsh, 1983*. They confirmed that having a pseudo-substrate where the P-site in PKI is an alanine, is essential for high affinity binding while the arginines that precede the P-site, as well as the *P*+1 hydrophobic residue, are also important for binding of both substrates and pseudosubstrates. To achieve high-affinity binding, however, hydrophobic residues that preceded the inhibitor site were required. Based on their peptide studies and subsequent biophysical studies, Walsh and his colleagues predicted that this hydrophobic motif was an amphipathic helix (*Glass et al., 1989a*; *Glass et al., 1989b*; *Reed et al., 1989*). Localization of the inhibitory region to the N-terminus of PKI and the rigorous characterization of the inhibitor peptide (IP20, residues 5–24) enabled crystallization of that first PKA structure bound to IP20 (*Knighton et al., 1991a*). A second paper (*Knighton et al., 1991b*), which completely validated the peptide predictions, showed precisely how IP20 bound with high affinity through basic residues near the active site and a distal amphipathic helix (*Figure 5*). Showing how both charged and hydrophobic residues contribute to peptide binding at sites that are distal to the phosphoryl transfer site did not explain, however, how the synergistic high-affinity binding of ATP and IP20 was achieved. The RIα subunit is also a pseudosubstrate and displays the same synergistic high affinity binding with ATP (*Herberg and Taylor, 1993*; *Herberg et al., 1999*). Whether other kinases have such pseudosubstrate inhibitors that display this synergistic high-affinity

binding with ATP is not clear, although two of the most important biological inhibitors of the PKA C-subunit, PKI and RIα, show this property, while RII subunits of PKA are actually substrates and not pseudo-substrates. The difference between substrate inhibitors and pseudo-substrate inhibitors is fundamental. Do other kinases have physiologically relevant pseudo-substrate inhibitors that binding with high affinity?

The specific features that convey high affinity binding to PKI are the amphipathic helix that docks into a hydrophobic groove on the C-Lobe while the high-affinity binding of RIα is due to the first CNB domain (*Kim et al., 2005*; *Kim et al., 2007*), CNB-A, which docks onto the Activation loop and the αG-helix. These docking sites can be thought of as substrate tethering sites. While ATP binds with a 15 μM affinity (Km) when phosphorylating a small peptide, such as Kemptide (G-R-R-G-S-L), in the presence of PKI or IP20 (residues 5–25) it binds with an affinity of ~60 nM. Similarly, IP20 on its own has an affinity of ~200 nM, while in the presence of MgATP, the affinity is ~1 nM. This high-affinity synergistic binding also requires two $Mg^{2+}$ ions that bind to the phosphates of ATP and neutralize the negative charge of the phosphate (*Herberg et al., 1999*), which allows the N- and C-lobes to fully close. In the fully closed conformation F54 in the G-loop is adjacent to F187 that follows the DFG motif. In this structure the γ-PO4 is well shielded from water, a phenomenon which is thought to facilitate the transfer of the phosphate (*Figure 5*). The second $Mg^{2+}$ ion, in particular, bridges the D184 in the C-Lobe with K72 in the N-Lobe allowing for full closure of the catalytic cleft. These unique features of IP20 allowed us to trap the fully closed conformation in our first structures of the PKA catalytic subunit (*Zheng et al., 1993*).

## Capturing dynamics

Identifying allosteric sites that lie distal to the catalytic site where phosphoryl transfer takes place is one of the most important challenges facing the kinase signaling community, as these sites which may be only transiently sampled, can nevertheless be excellent therapeutic targets. While crystal structures and cryoEM structures provide us with high-resolution portraits of folded proteins, these structures are static snapshots. In contrast, NMR provides a residue-specific window into dynamics. NMR studies of the PKA C-subunit, first described three states that are associated with the fully phosphorylated protein - uncommitted, committed, and quenched states (*Masterson et al., 2011*). It should be emphasized that these states are all active conformations where the R-spine is intact; they simply correlate with the opening and closing of the active site cleft. The apo state, which is in an open conformation, is uncommitted to catalysis while the binding of nucleotide, which completes the C-Spine, commits the kinase to catalysis. The intermediate states with bound nucleotides correspond to partially closed states where the C-terminal tail and the G-loop are still dynamic. How these states correlate with the 'Communities' of the kinase core was demonstrated by McClendon, et al (*McClendon et al., 2014*). The quenched and fully closed conformational state is captured when both ATP and the pseudo-substrate, IP20, bind with high affinity. While these initial NMR studies captured the backbone dynamics, it is the side chains that report much of the entropy-driven dynamics (*Kim et al., 2017*), and to explore this hydrophobic space required labeling the side chains. When side chains (Val, Leu, and Ile) were labeled, one could for the first time observe the correlated motions of the hydrophobic core architecture, which experimentally captured the entropy-driven allostery (*Ahuja et al., 2019a*).

Defining and targeting the allosteric sites of protein kinases is a holy grail, and if we delve into the architecture of the kinase core as well as the tails and domains that flank the kinase core in the PKA C-subunit, there are many sites where the core can become 'uncoupled' from the flanking motifs. Within the core, many sites lie distal to the active site that can uncouple ATP and peptide/protein binding. Which of these mechanisms can abolish the synergistic high-affinity binding of ATP and IP20 or can all of them do it? Some of these 'uncoupling motifs' such as F327, Y204, and E230 are sites that we have experimentally queried, whereas others such as W196R, L205R, and E31V are known disease mutations that correlate with Cushing's Syndrome (*Omar et al., 2022*; *Omar et al., 2023*; *Walker et al., 2021*; *Olivieri et al., 2022*; *Walker et al., 2019*).

E230 is a critical residue for recognizing the P-2 arginine in PKA substrates. It couples the peptide to Y204 in the *P*+1 loop and to E170 in the Catalytic loop. The E230Q mutant uncouples the N- and C-lobes by 'freezing' the enzyme into a stable open conformation that can no longer bind ATP (*Wu et al., 2005*). Y204 is an example of an allosteric site that is not associated with a change in

conformation following replacement with Ala. Functionally this mutation reflects instead an inability to transfer the γ-phosphate of ATP to a protein substrate while it is still fully capable of transferring the γPO$_4$ to water. Even though there is not a change in structure in the Y204A mutant, mechanistically the Y204A mutation reflects a change in dynamics that can lead to significant changes in the communities that the residue interacts with. Although changes in dynamics can be captured and validated by NMR, changes in dynamics that lead to changes in stability can also be predicted by LSP alignment. LSP alignment identified this mutation, Y204A, as part of a dynamics-driven allostery mechanism that can be described as a 'Violin' model that draws an analogy between the distribution of thermal vibrations in proteins to vibrational patterns in a violin (*Kornev and Taylor, 2015*; *Ahuja et al., 2017*; *Ahuja et al., 2019b*; *Madan et al., 2023*).

F327, which lies outside the kinase core, highlights another unique feature of PKA that is conserved in all the AGC kinases (*Kannan et al., 2007*). F327 is part of a C-terminal tail that wraps around both lobes of the kinase core and is an essential feature of the ATP binding site (*Figure 5*). When all of the residues comprising the hydrophobic shell around the adenine ring of ATP were mutated to alanine, and analyzed in a yeast screen, only two were found to block viability, F327 and L173 (*Kennedy et al., 2009*). We now know that L173 is a C-spine residue, but F327 lies outside of the kinase core. Nevertheless, it is absolutely essential for the binding of ATP. Replacing F327 with alanine reduces the Km for ATP over 10-fold (~450 μM) and completely abolishes the high-affinity binding of ATP.

Three Cushing syndrome mutations were also shown to 'uncouple' the high-affinity binding of inhibitor proteins. L205R uncouples binding of both RI and RII subunits as well as PKI by disrupting binding to the *P*+1 pocket (*Walker et al., 2019*), while W196R disrupts binding to the cyclic nucleotide-binding domain (CNB-A) of both RI and RII subunits (*Omar et al., 2022*; *Gibson and Taylor, 1997*). E31V is an example of a mutant that uncouples the αA-Helix from the Activation loop, and this mutation also abolishes the synergistic high-affinity binding of ATP (*Walker et al., 2021*). Paul Herman identified a set of mutants on the C-lobe of the C-subunit that selectively enhanced the docking of protein substrates (*Deminoff et al., 2006*; *Deminoff et al., 2009*). He defined these as tethering sites, and all of these are directly linked to the αF/αE-Helices. So, there are a number of ways that one can uncouple the synergistic high-affinity binding of ATP and IP20 or RIα by mutating sites that are distal to the catalytic site.

## F100A mutation highlights the importance of the αC-β4 Loop

Another mutation that lies in the αC-β4 loop, F100A, was recently described (*Deminoff et al., 2006*; *Deminoff et al., 2009*). Adjacent residues in the αC-β4 loops (V104I, L103F/I, and P101A) are actually oncogenic mutations in several protein kinases, including PKA although none of these have been validated. The synergistic binding of ATP and IP20 was completely abolished for the F100A mutant; even though the Kms for Kemptide and ATP for the F100A mutant were similar to the wild type (wt) C-subunit (*Olivieri et al., 2023*). Peptide assays are a good way to evaluate the kinetic machinery of a protein kinase, and biochemically, based on the Kemptide peptide assay, the kinetic mechanism for the C-subunit was not altered significantly by the F100A mutation. However, the peptide assay does not reflect how a protein kinase works in cells where it phosphorylates other proteins, not peptides. One will never replicate Michaelis Menton kinetics in cells where the assay depends on having a large excess of a small peptide substrate. This is, of course, relevant for peptides or hormones binding to a receptor on the surface of the cell, but inside cells, the protein substrates are not typically in a huge excess of the kinase and the interactions are not diffusion-limited. They are instead dependent on co-localization where the protein substrate is tethered in close proximity to the kinase. This can be achieved by tethering directly to sites such as those described by Paul Herman on the C-Lobe of the PKA kinase core (*Deminoff et al., 2006*; *Deminoff et al., 2009*) or by binding to a kinase scaffold protein in a way that brings the P-site close to the active site of the kinase (*Pawson and Scott, 1997*). With this mechanism protein phosphorylation is not diffusion-limited; however, it still highly depends on Brownian Dynamics (BD) (*Huber and McCammon, 2019*; *Northrup et al., 1984*); the charged residues are now simply tethered in close proximity to the active site.

The importance of the αC-β4 loop was actually initially hinted at by computational analyses that involved Markov State modeling and NMR-restrained replica-averaged metadynamics (RAM). These analyses suggested that there may be a flip of the αC-β4 loop in the apo protein and in the mutant (*Olivieri et al., 2023*). As seen in *Figure 4*, no such flip has been observed so far in any of the crystal

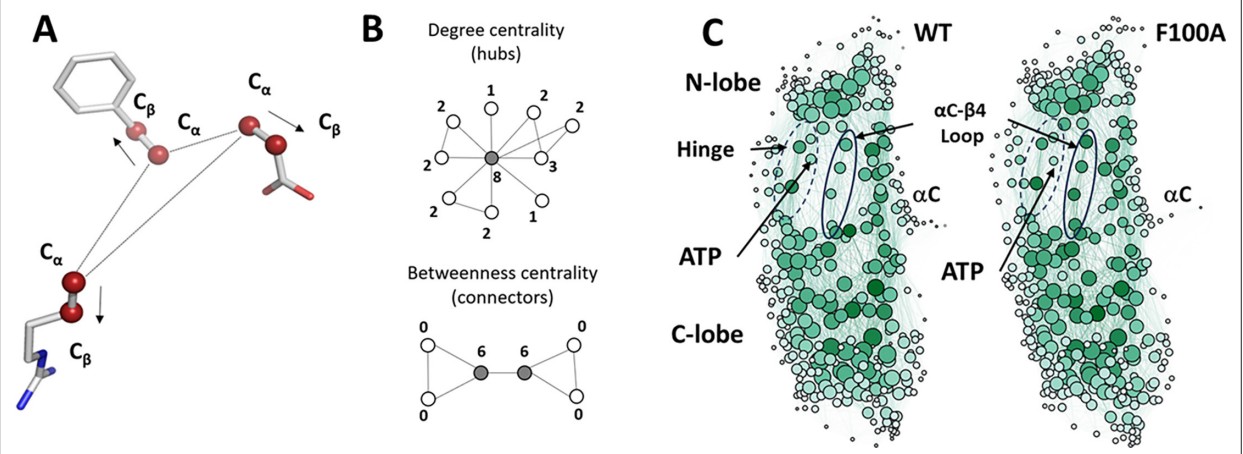

**Figure 6.** Local spatial pattern (LSP) alignment method. (**A**). Spatial patterns detected by the LSP-alignment are formed by Cα-Cβ vectors. Both mutual positions and orientation in space of these vectors are taken into account. (**B**). Two major centralities that characterize graphs. Degree centrality (DC) is the sum of connections for each node. The highlighted node in the middle has the highest DC value of 8. Residues with high level of DC are local 'hubs.' Betweenness Centrality (BC) of a node is the number of shortest paths between all other pairs of nodes that pass through this node. Two highlighted nodes are the main connectors in the graph with BC equal to 6. Residues with high level of BC are global connectors between local hubs. (**C**) Visualization of protein residue networks (PRNs) of protein kinase (PKA) before and after the mutation laid out by Gephi software package using ForceAtlase2 algorithm. The diameter of nodes is proportional to the residues DC. Nodes with higher BC have darker color. Four residues from theαC-β4 loop (103-106) are highlighted by oval. The Hinge region is indicated by the dashed oval.

structures of the C-subunit to date (wt, E230Q, unphosphorylated C, and the apo C-subunit), regardless of closed or open conformation. The subsequent NMR studies of the F100A mutant also highlighted V104 and I150 as critical residues. I150 is a critical part of the hydrophobic core architecture of the C-Lobe that anchors the αE Helix to the C-spine (L172 and I174 in β7) and to L167and P169 in the Catalytic loop (see above discussion and *Figure 2*). In contrast, V104, which is part of the αC-b4 loop, is one of three shell residues (SH1, SH2, and SH3) that flank the R-spine residues in the N-lobe and V104 also touches ATP (*Meharena et al., 2013*). Specifically, one of the methyl side chains of V104 touches the adenine ring of ATP so it is one of the N-lobe adenine capping residues for ATP (*Kennedy et al., 2009*; *Figure 5*). All of the adenine capping residues in the C-subunit come from the N-Lobe and the C-tail except for L173, which is the only capping residue that comes from the C-Lobe. The C-spine is so important, because it links all of the hydrophobic capping machinery in the N-Lobe with the hydrophobic core architecture in the C-Lobe. This residue is not always conserved as a valine, but it is typically a small hydrophobic residue. This hydrophobic residue also touches the well-studied gate-keeper residue, which in PKA is M120; the gatekeeper is another shell residue (SH3) (*Poulikakos et al., 2010*; *Garske et al., 2011*).

The long computational studies (2 μsec) carried out by Veglia predicted that the αC-β4 loop undergoes a flip in the apo state, while the ITC results showed that the F100A mutant is also more stable than the wt C-subunit. Most importantly, the biochemical studies clearly demonstrate that this mutation disrupts the synergistic binding of ATP and IP20. We thus asked whether LSP could identify changes in dynamics that correlate with the NMR results. Could the LSP analysis, which requires relatively short simulation times, predict residues or regions in the C-subunit that could account for enhanced or reduced stability that results from this mutation even if they cannot predict a conformational change?

## Comparing the LSP alignment of F100A with wt C-subunit

LSP alignment is a computational method developed in our laboratory for capturing conserved patterns formed by Cα-Cβ vectors in proteins (*Figure 6A*). Initially, it was utilized to identify conserved hydrophobic ensembles in protein kinases (*Kornev et al., 2006*; *Kornev et al., 2008*). More recently, this technique was applied to MD simulations, in an effort to analyze stable regions in Protein Kinase A (*Kornev et al., 2022*). By comparing spatial patterns formed by Cα-Cβ vectors in differing conformations generated via MD simulation it is possible to analyze thermal vibrations of residues. These

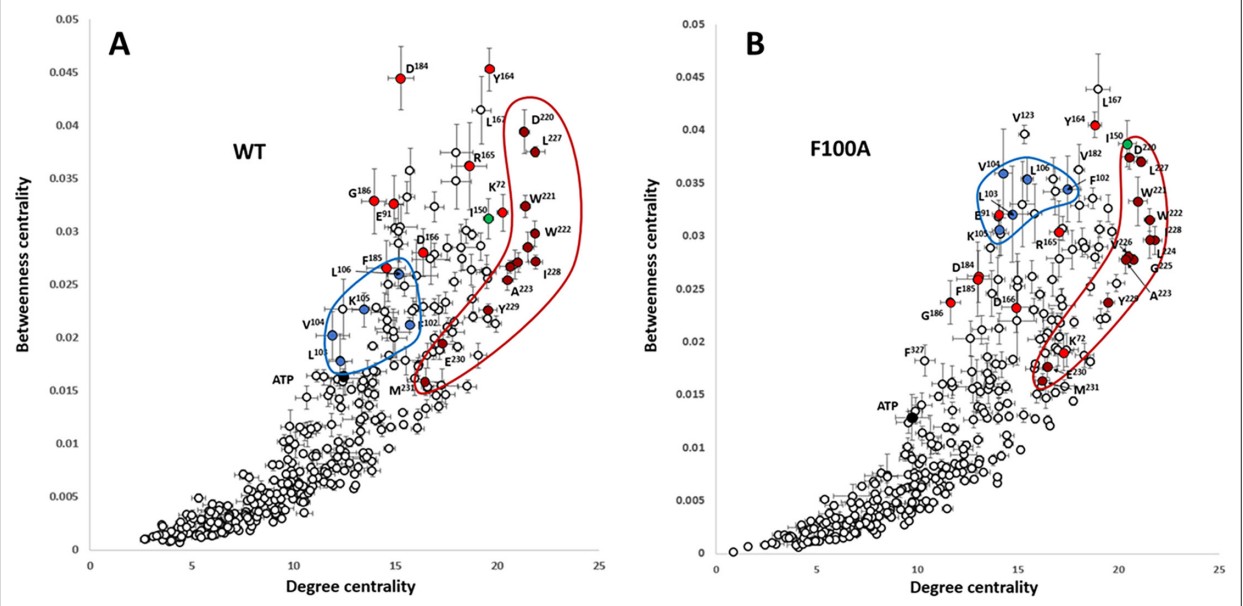

**Figure 7.** Distribution of Betweenness centrality vs.Degree centrality for protein kinase (PKA) residues: wt C-subunit (**A**) and F100A (**B**). Residues with Degree centrality (DC) are local 'hubs,' representing the most stable parts of the molecule. Residues with high Betweenness centrality (BC) are global connectors and are 'bottlenecks' between densely interconnected 'hubs.' Residues of the central αF-helix are shown as brown circles. Their high levels of DC and BC don't change upon the F100A mutation, meaning that the helix remains to be a central structural element in the PKA mutant. Five residues from the αC-β4 loop are highlighted as blue dots. The significant increase in BC and a certain increase in DC shows that in the mutant this group of residues becomes a significant point of connectivity between the kinase lobes. On the contrary, the highly conserved residues highlighted as red circles show a drastic decrease in BC. Error bars represent standard error calculated for five 10 ns trajectories.

motions occur on a sub-nanosecond timescale and are considered to be the foundation of dynamics-driven allosteric effects, which were predicted by *Cooper and Dryden, 1984* and have been observed in multiple proteins (*Popovych et al., 2006*; *Petit et al., 2009*).

LSP-alignment is a graph-theory-based method that implements a Protein Residue Network (PRN) approach (*Di Paola and Giuliani, 2015*). As we demonstrated, two major centralities of such PRNs can contain important information on the local stability (Degree centrality, DC) and global connectivity (Betweenness centrality, BC) of the protein (*Kornev et al., 2022*; *Figure 6B*). Our purpose here was to identify changes in LSP-based PRNs associated with the F100A mutation, specifically assessing whether these changes relate to the dynamic features of the αC-β4 loop and correlate with the NMR results. *Figure 6C* shows the corresponding PRNs laid out by the ForceAltlas2 algorithm (*Jacomy et al., 2014*). This algorithm treats the weights assigned to edges as attractive forces that balance the imposed repulsion of the nodes. In LSP-based PRNs, compact and highly interconnected nodes correspond to more ordered regions of the protein where Cα:Cβ vectors move cohesively and preserve their mutual orientations. Both PRNs showed very similar general structures, featuring dense groups of residues that correlate with the N and C-lobes. These are linked by the Hinge, αC-β4 loop and αC-helix. The analysis also includes the ATP molecule, that was described by the N1-C8 vector. Notably, the darker color of the αC-β4 loop residues in the F100A mutant indicates the increased level of BC.

To analyze changes of BC and DC in more detail we plotted them on scatterplots (*Figure 7*). Residues from the αF-helix have the highest levels of DC in both the wt C-subunit and F100A. This is consistent with the fact that this helix is the most stable structural element of the kinase core and is known to be a major scaffold for the catalytic machinery of these enzymes (*Figure 1*; *Kornev et al., 2008*). Several of the αF helix residues such as D220, W221, W222, and L227 from the C-spine, score high for both DC and BC. This indicates that these residues act both as hubs and connectors. The most significant change in the F100A mutant is the increase of BC for the five residues from the αC-β4 loop (highlighted in blue). Conversely, a set of highly conserved residues (marked in red), previously noted for having the highest levels of BC *Kornev et al., 2022*, show a reduction in their DC, and this reduction is even more pronounced in the BC values in the F100A mutant.

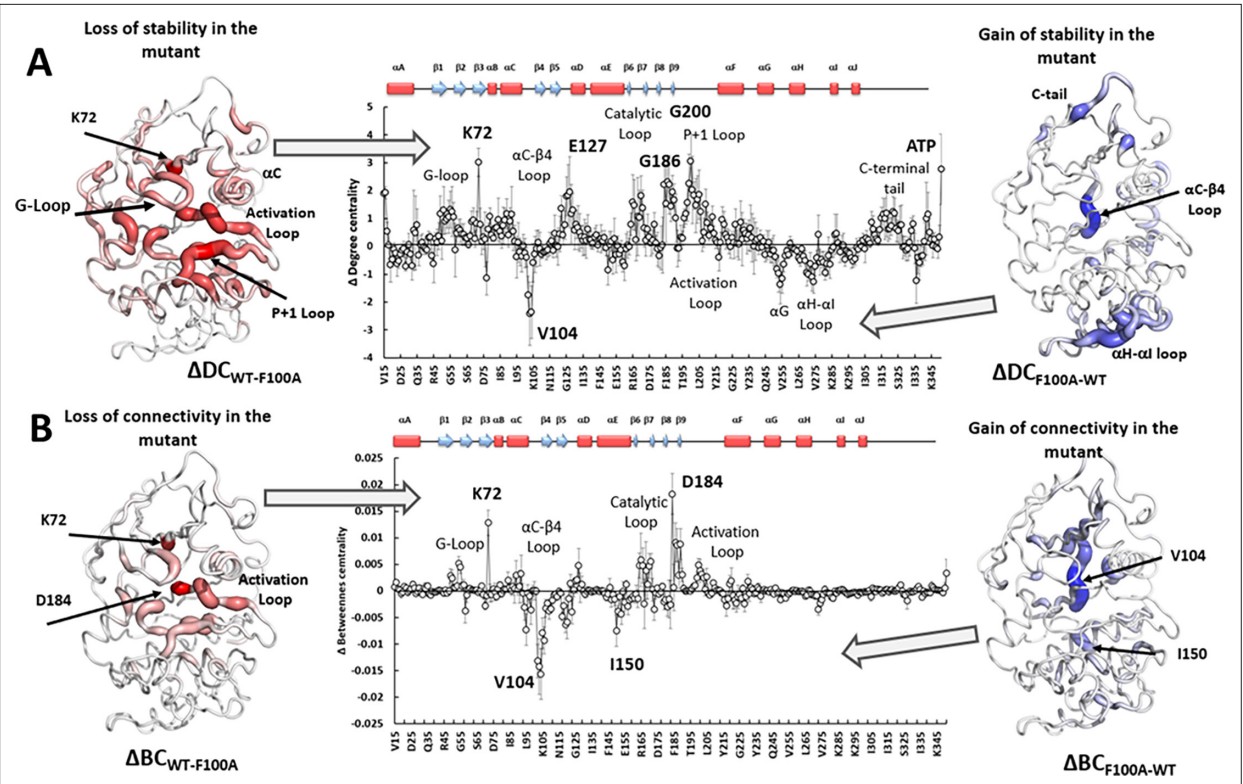

**Figure 8.** Changes of Degree centrality (DC) and Betweenness centrality (BC) in protein kinase (PKA) upon F100A mutation. The middle graphs represent changes in the corresponding parameters. Positive changes are mapped on the PKA structure (left) with dark red color corresponding to the maximum changes. On the right the negative values of the changes are mapped on the structure to illustrate their distribution. Dark blue color corresponds to the most negative values. Error bars represent standard error calculated for five 10 ns trajectories. The secondary structure of PKA is shown on top of the sequence axis for reference.

The detailed values of DC and BC changes associated with the F100A mutation are shown in *Figure 8*. Since both of these parameters have positive and negative values, we visualize them separately, mapping them onto the PKA structure. Positive changes in DC correspond to areas that lose the stability of their Cα-Cβ vectors upon the mutation (*Figure 8A*, left, red). These include the catalytic area of the kinase, and most importantly, universally conserved K72, the DFG motif, including D184, and the substrate binding site. Notably, the ΔDC value for ATP is one of the highest (*Figure 8A*, center), indicating a significant loss of stability of the adenine ring in the mutant. Negative changes in ΔDC (*Figure 8A*, right, blue) correspond to areas of PKA that become more stable in the mutant. Significant changes were primarily localized in the αC-β4 loop. Similarly, negative changes in ΔBC values (*Figure 8B*, right, blue) were also found in the αC-β4 loop, signifying that the role of the major connecting area between the lobes shifts from the catalytic area (*Figure 8B*, left, red) to this region. Overall, the F100A mutation causes the rigidification of the αC-β4 loop, especially V104 at the tip becomes more stable and rigid following the mutation of F100. Our analysis suggests that the major connectors between the N- and C-lobes such as K72 from β3 stand (N-lobe) and D184 from the DFG motif (C-lobe) in the wt protein get impaired and destabilized in response to the F100A mutation. The rigidified αC-β4 loop loses the dynamic coupling between the two lobes and the connectivity between both lobes changes, as seen by V104 and I150, which become the major connector points following the mutation. Overall, the mutation at the αC-β4 loop changes the allosteric communication between the two lobes.

In general, these results can be interpreted as a disruption of dynamic communication between the two lobes where ATP binding in the N-Lobe comes together with the major catalytic machinery in the C-Lobe (*Figure 2*). The prominent role of D184 in the C-lobe and K72 in the N-Lobe, hallmark features of the conserved regulatory triad, are also altered. Upon the mutation, a robustly stable region around

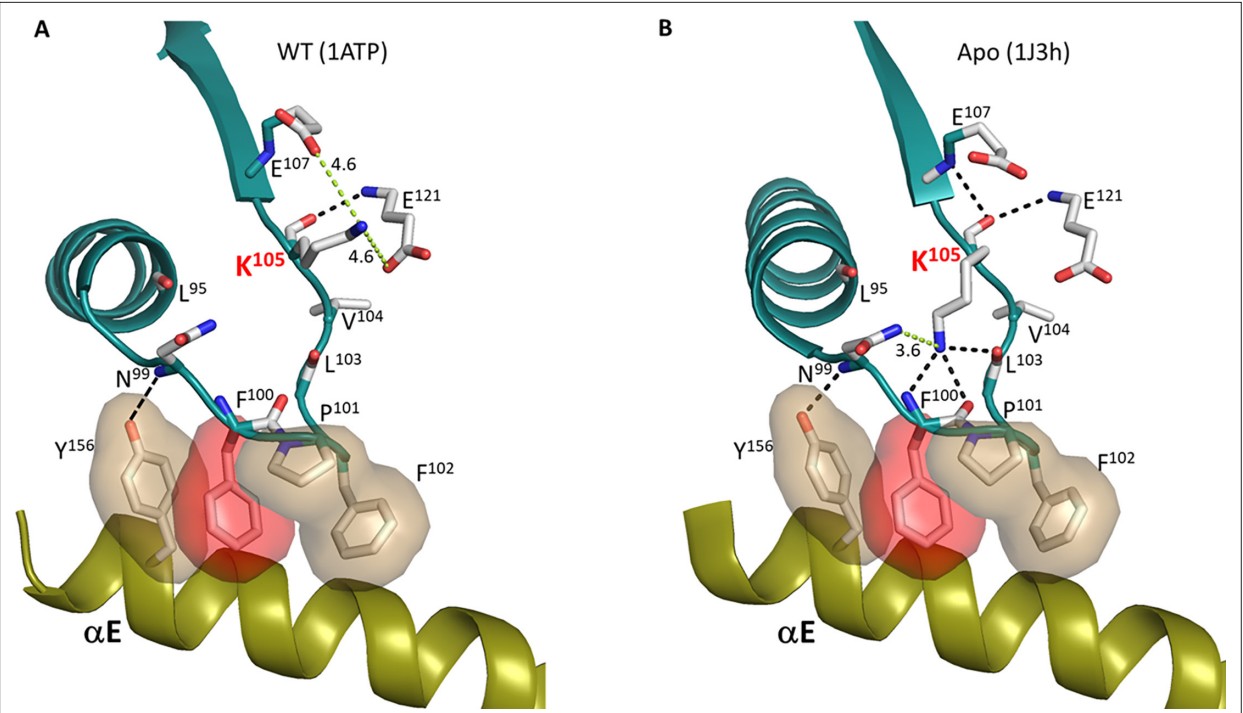

**Figure 9.** Dynamic feature of K105 on αC-β4 loop. (**A**). In the wt PKA:ATP:Mg:PKI complex structure, the side chain of K105 is likely interact with E107 and E121, and its main chain carbonyl forms a H-bond to backbone amine of E121. (**B**). In the wt Apo protein kinase (PKA) structure, the K105 side chain interacts with β-turn, whereas its main chain carbonyl still H-bonds to backbone amines of E107 and E121. The H-bonds are shown as dash line in black, and the non H-bond are colored in green and the distances are labeled.

F102-L106 becomes a dominant connector between the lobes, leading to a significant disruption of correlated dynamics observable in the active site of the wt C-subunit.

## Capturing side chain dynamics

Our initial preliminary LSP alignment comparison of wt C-subunit in the presence of ATP with the F100A mutant indicated significant destabilization of the entire catalytic machinery in the C-lobe in parallel with enhanced stability of the αC-β4 loop, a node in the C-tail and the αH-αI loop (*Figure 8A*). While a much more extended analysis is needed to validate the LSP predictions, the overall results clearly show that the dynamic features of the αC-β4 loops have changed in the F100A mutant. As discussed earlier and as indicated in *Figure 4*, we did not see significant changes in the αC-β4 loops when we compared our various crystal structures, which included open and closed conformations of the active kinase as well as an inactive unphosphorylated C-subunit. Specifically, the tip of the αC-β4 loops remained anchored to the αE Helix in all of these structures. Intrigued by our LSP results with the F100A mutant, we thus looked more carefully at the side chain residues in the αC-β4 loop in our various structures. We asked specifically if there were differences in any of the side chain residues that could be predictive of subsequent conformational changes in the backbone as was predicted by the NMR studies. The side chains of the αC-β4 loop in the ternary complex with ATP and IP20 are shown in *Figure 9A*, and no significant differences were seen for the E230Q mutant and the unphosphorylated C-subunit. In this structure, the side chain of K105 was toggling (~4.6 A) between two carboxyl groups (E107 and E121), and its backbone carbonyl (the first residue of β4) is hydrogen bonded to the backbone amide of E121 (the last residue of β5) (*Figure 9B*). While the apo protein showed no major changes in the backbone of the αC-β4 loop, the side chain of K105 has flipped and is now interacting with carbonyls in the β-turn residues. It is also close to the side chain of N99. Based on the crystal structures, this space is not sampled by K105 in the wt C-subunit; instead, this space is filled by ordered water molecules.

To look more carefully at the space that is sampled by the side chain of K105, we looked at the MD simulations that were used for the LSP analysis of the wt binary complex of the C-subunit and the

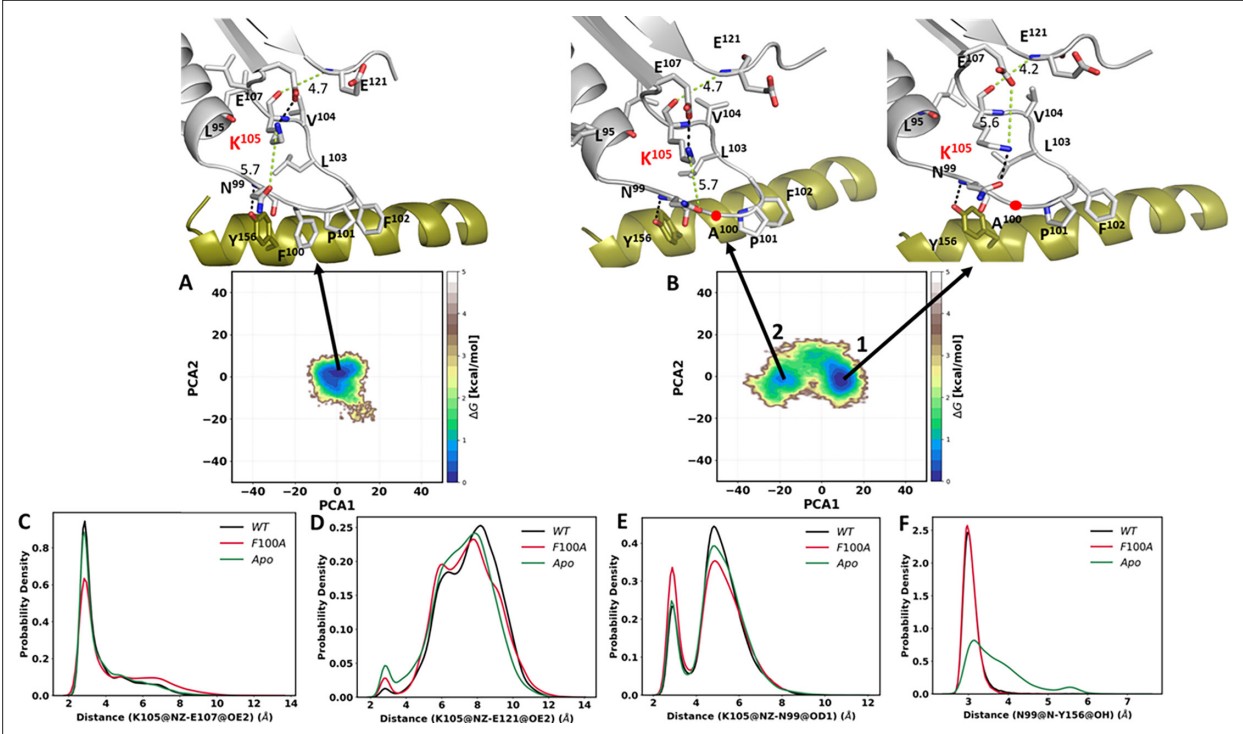

**Figure 10.** Energy landscapes of wt C-subunit and F100A mutant in binary complex. The free energy landscapes (FEL) were generated based on the principal component analysis (PCA) for the wt C-subunit (**A**) and F100A mutant (**B**). The H-bonds are shown in the black dash line and the non H-bond in green along with the distance in Angstroms (Å). The probability distribution plots for specific side chain dynamics are also shown; (**C**) K105 NZ to E107 OE2, (**D**) K105 NZ to E121 OE2, (**E**) K105 NZ to N99 OD1, and (**F**) N99 N to Y156 OH. The probability density function (PDF) is a relative measure of how densely data points are distributed along the x-axis.

F100A mutant. As seen in *Figure 10A*, for the wt C-subunit there is one global energy minimum, and the predominant structure for the global energy minimum correlates well with our structure of the ternary complex (*Figure 9A*). Based on the simulations, however, instead of toggling between E107 and E121, the side chain of K105 interacts predominantly with the side chain of E107 (*Figure 10C*), while it rarely samples the side chain of E121 (*Figure 10D*). In contrast to the wt C-subunit and the apo protein, the mutant shows two energy basins (*Figure 10B*). The minor local minimum corresponds closely to the wt C-subunit with K105 interacting with E107. In contrast, the major global energy minimum shows the side chain of E105 interacting with the side chain oxygen of N99 and far from the side chain of E107. The probability distribution of this interaction with N99 is shown in *Figure 10E*, which confirms that the mutant has a stronger propensity to interact with N99. As discussed above, N99 is thought to be a critical residue based on the importance of the homologous residue in BRaf (R509) for dimerization and because the backbone of this residue is always anchored to a key residue in the αE Helix (Y156) in every kinase. To interrogate the strength of the Y156 hydrogen bond to the backbone amide of N99, we also looked at its probability distribution in the simulations (*Figure 10F*). While the H-bond is strong in the complex and in the mutant, it is surprisingly destabilized in the apo structure.

After investigating the binary complex of the C-subunit for both wt and F100A, we next analyzed the ternary complex. Using the MD simulations, we asked specifically if the F100A mutation alters the dynamic properties of K105 when both ATP and PKI are present. The free energy landscape (*Figure 11*) depicts the overall energy profile for the wt ternary complex and the F100A mutant. The wt ternary complex, in general, behaves similarly to the binary complex. Both exhibit a single global energy minimum (*Figure 11A*). In contrast, the F100A mutation shows enhanced dynamics for both binary and ternary complexes, although there are multiple energy basins in the ternary complex (*Figure 11B*). The side chain dynamics of K105 also show different interaction profiles (*Figure 11C, D and E*), and differences are seen even when just the binary and ternary complexes are compared. In the wt binary complex, the side chain of K105 predominately interacts with the side chain of E107

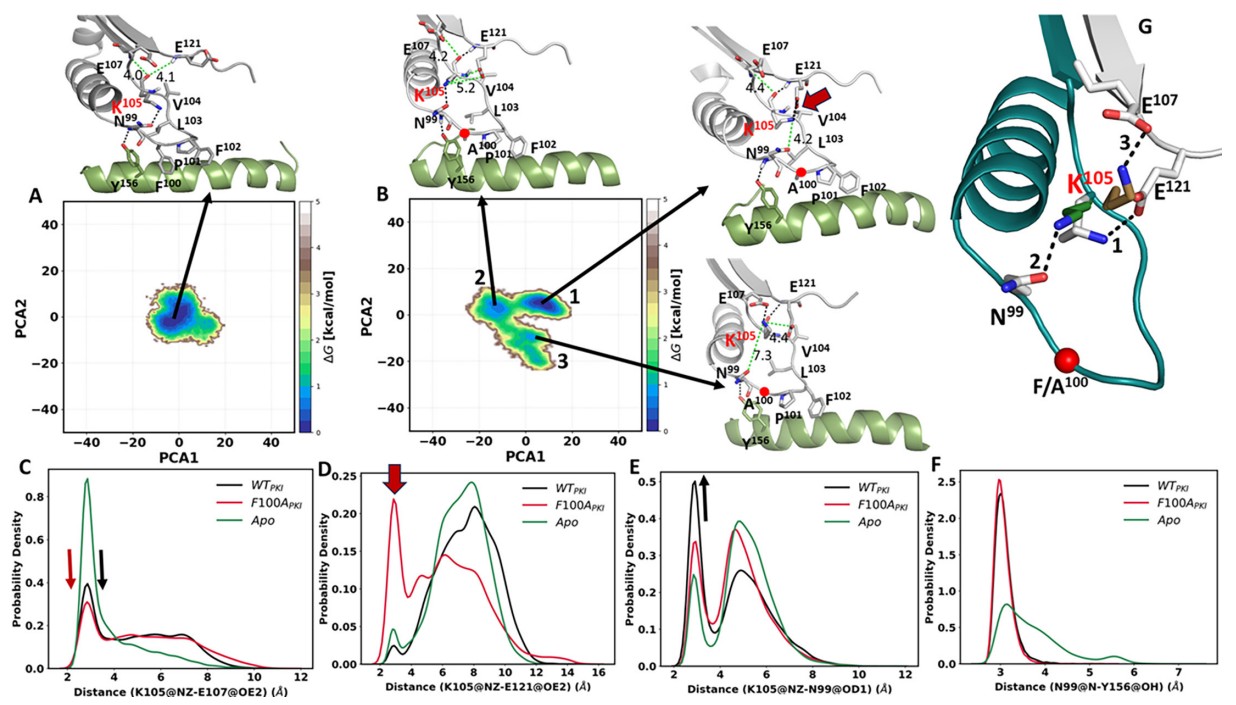

**Figure 11.** Energy landscapes of wt C-subunit and F100A mutant in ternary complex. The free energy landscapes (FEL) were generated based on the principal component analysis (PCA) for the wt C-subunit with PKI (**A**) and F100A mutant with PKI (**B**). The H-bonds are shown in the black dash line and the non H-bond in green along with the distance in Angstroms (Å). The probability distribution plots for specific side chain dynamics are also shown; (**C**) K105 NZ to E107 OE2, (**D**) K105 NZ to E121OE2, (**E**) K105 NZ to N99 OD1, and (**F**). N99 N to Y156 OH. (**G**) The enhanced side chain dynamics of K105 in the F100A ternary complex: the side chain of K105 toggles to its neighboring residues, E121 (white, global energy minima), N99 (green, first secondary minima), and E107 (sand, second secondary minima). The probability density function (PDF) is a relative measure of how densely data points are distributed along the x-axis.

(*Figure 10C*) and interacts more transiently with the side chain of N99 (*Figure 10E*). This is in contrast to the wt ternary complex where the interactions of K105 and N99 side chains increase (*Figure 11E*), while the interaction with E107 decreases (*Figure 11C*). The side chain of K105 in the wt binary, ternary, and apo states rarely samples the side chain of E121 (*Figures 10D and 11D*). The hydrogen bond between Y156 and N99 is also maintained in the binary and ternary complexes, but broken in the apo protein (*Figures 10F and 11F*).

Does the F100A mutation introduce additional changes in the crosstalk of the ternary complex? The major difference is between K105 and E121, which correlates with the primary energy minimum (*Figure 11D*). This interaction is not seen in any of the other complexes. In addition, the side chain of K105 has a reduced propensity for interacting with the side chain of E107 as shown in the second local minimum. Finally, there are no significant changes between Y156 and the backbone amide of N99 in the mutant (*Figure 11F*). Hence, the enhanced side chain dynamics of K105 have been observed following the F100A mutation where the side of K105 toggles between E121 and N99, and to some extent E107 (*Figure 11G*). In general, these results agree with the NMR experimental results and suggest that the F100A mutation alters the dynamic interactions of the αC-β4 loop, leading to the hypothesis of the disruption in the allosteric communication between both lobes.

The most striking difference, based on the MD simulations, is the enhanced interaction of K105 with the side chain of E121, a key residue that binds directly to the adenine ring of ATP. As seen in *Figure 12A*, in the crystal structure of the ternary complex the backbone amide of E121 hydrogen bonds to the backbone carbonyl of K105, while the backbone carbonyl of E121 hydrogen bonds directly to the adenine ring of ATP. The hydrophobic packing between the ATP adenine ring and V104, a shell residue, is extended to the R-spine residues, L95 (RS3) and L106 (RS4), which is also highlighted. How these predicted changes in dynamics in the F100A mutant correlate with enhanced stability and/or altered function needs to now be validated experimentally.

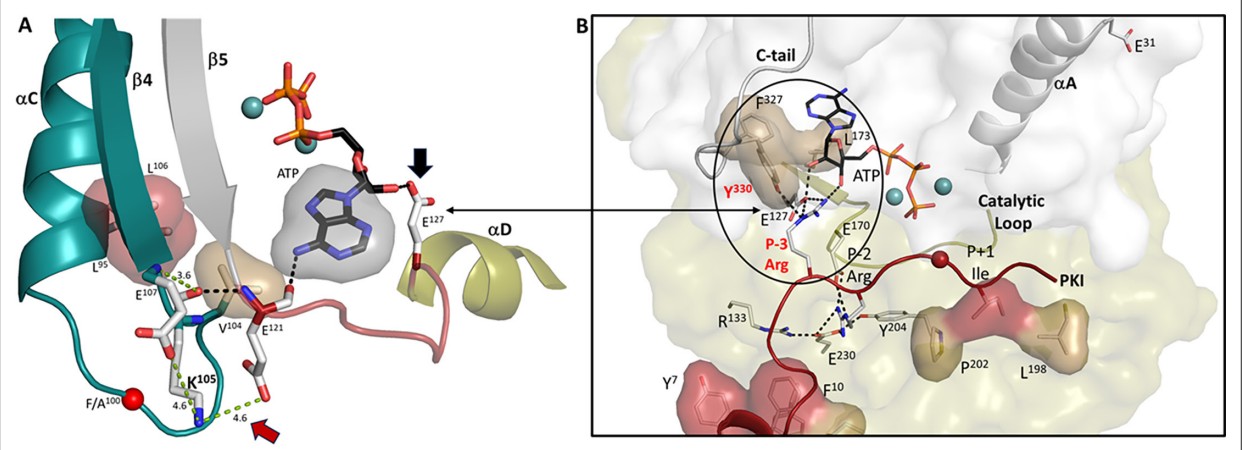

**Figure 12.** High-affinity binding of adenosine triphosphate (ATP) and protein kinase inhibitor (PKI) converge at the αC-β4 loop. (**A**). V104 inαC-β4 loop (in teal) is hydrophobically anchored to the adenine ring of ATP. The main chain carbonyl of K105 forms an H-bond to the backbone amide of E121, while its main chain carbonyl hydrogen bonds to ATP. The distance between K105 and E121 side chains is strengthened in the F100A mutant. Two spine residues, L95 and L106, in this loop are also shown. The linker that joins the N- and C-lobes (red) is flanked by E121 and E127. (**B**). Hydrophobic capping of the adenine ring of ATP is mediated mostly by N-lobe residues including V104 in the αC-β4 loop, as well as F327 in the C-terminal tail. In contrast, the P-3 to *P*+1 peptide is anchored to the catalytic machinery of the C-lobe. By binding to L173 in the C-lobe, the adenine ring completes the C-Spine, and thus fuses the adenine capping motif in the N-lobe with the extensive hydrophobic core architecture of the C-lobe. In the fully closed conformation, the side chain of Y330, also in the C-terminal tail, is anchored to the ribose ring of ATP. The only direct contact of the peptide/catalytic machinery with the N-lobe is mediated by the P-3 arginine which binds to the ribose ring of ATP and to E127 in the linker. In the fully closed conformation, the P-3 arginine also binds to the side chain of Y330 in the C-terminal tail. Some of the mutations that disrupt the synergistic high-affinity binding of ATP and peptide/protein (E230Q, Y204A, F327A, L173A, and E31V) are highlighted. The hydrophobic residues in the amphipathic helix and *P*+1 inhibitor site of PKI are shown in red.

## Summary and future directions

Based on the NMR analysis of the F100A mutant, we more rigorously examined the αC-β4 loop showing first how it is firmly anchored to the hydrophobic core architecture of the C-lobe. In addition, we reviewed the synergistic high-affinity binding of ATP and PKI, a phenomenon that was described in detail many decades ago for PKI but never fully explained in terms of its mechanism. In the F100A mutant, this synergy is uncoupled. Finally, given that both NMR and the Markov State model suggested changes in the αC-β4 loop, we carried out an LSP analysis of the mutant. The LSP analysis correlated remarkably well with the NMR predictions. To explain this correlation, we compared various crystal structures of the C-subunit to see if we could detect differences in side chain dynamics since the backbone did not change significantly. LSP alignment is a tool that is perfectly poised to detect alterations in side chain dynamics as it identifies differences in the geometry of the α carbon/β carbon vectors (*Figure 6*), a property that likely precedes any conformational changes, which take a much longer time to simulate with classical Molecular Dynamics. MD simulations of the binary and ternary complexes of the wt C-subunit, the F100A mutant, and the apo C-subunit were also used to explore changes in side chain dynamics.

Although in these short MD simulations we did not detect a flip of any backbone residues in the αC-β4 loop, we did see significant differences in the side chain dynamics of K105 in the structure of the apo protein. This could, in principle, correlate with a weakening of the β-turn. Based on the MD simulations, the side chain of K105 is also flipped in the F100A mutant. In both cases, the side chain of K105 is exploring the space that is occupied by the ordered water molecules in the other structures. These water molecules play an essential role in stabilizing the β-turn motif. We thus predict that the enhanced flexibility of the side chain of K105 may be an initial step that leads to subsequent changes in the backbone dynamics. This prediction needs now to be validated with crystal structures to see if the mutation actually 'freezes' an altered conformation of the αC-β4 loop. The potential importance of both N99 and K105 in mediating changes in dynamics also needs to be experimentally validated biochemically, by NMR, and by further computational analyses. We predicted that N99 is important due to is conserved hydrogen bonding to Y156 and the conserved packing of the side chain of Y156

with hydrophobic residues in the β-turn. This, however, is the first potential indication of a direct role for K105. The graphs of the hydrogen bond between the side chain of Y156 and the backbone amide of N99 show that this bond is very stable in the binary and ternary complexes and in the mutant but weakened in the apo protein (*Figure 10E*), and this would also be consistent with the prediction that the β-turn may be more flexible in the absence of ligands but stabilized in a different way in the F100A mutant.

Overall, our work suggests that the αC-β4 loop should be carefully examined in all protein kinases as it is a focal point for linking ATP binding to peptide/protein binding and for opening and closing of the catalytic cleft. Our early structure captured the high-affinity binding of both ATP and IP20, a pseudo-substrate inhibitor, in a fully closed conformation (*Figure 12B*). In this conformation, the IP20 peptide is firmly anchored to the catalytic machinery of the C-lobe, while F327 and Y330 in the C-terminal tail are anchored to ATP. The side chain of the P-3 arginine in IP20 interacts directly with ATP, E127, and Y330 in the C-terminal tail, and all are close to the αC-β4 loop. This convergence of the P-3 arginine side chain with Y330 may explain the enhanced synergy that is seen when the PKA C-subunit interacts with its pseudo-substrate inhibitors, PKI and RIα, and understanding this synergy is a major future challenge. E127 binds to the other ribose hydroxyl moiety. E121 and E127 flank the linker that joins the N-and C-lobes (*Figure 12A*), and this linker is likely a critical feature for opening and closing of the active site cleft. K105 in the αC-β4 loop can obviously sense differences in the apo, binary, and ternary complexes, and can also sense mutations. This space now needs to be further explored. The αC-β4 loop is also a good potential therapeutic target as the specific residues such as N99 and K105 that are mechanistically important are different in all kinases in contrast to the conserved residues such as K72 and D184 that position the phosphates of ATP. The importance of αC-β4 loop is also reinforced because the cancer mutations are located here (*Ruan and Kannan, 2018*).

Finally, given the extremely close correlation of the LSP analysis with the NMR results and the Markov State model predictions, as well as our principal component analysis (PCA) of the wt and F100A mutant, we suggest that LSP could be used as a predictor of dynamics for any mutant or even for carrying out an alanine scan of any protein. These sites could then be examined experimentally. Although purifying these proteins and especially labeling the hydrophobic side chains is not only a major time investment but also very expensive, these methods are essential to experimentally validate the importance of hydrophobic residues in mediating entropy-driven allostery. In contrast, LSP is rapid and relatively inexpensive and could easily be used as an initial screen to identify potentially important residues that contribute to dynamics.

# Materials and methods
## MD simulation

The catalytic subunit of PKA was prepared using the crystal structure (PDB: 1ATP) (*Zheng et al., 1993*) for all-atom MD simulations using the AMBER16 suite (*Case et al., 2005*). In order to study the allostery dynamics of the aC-b4 loop and compare the effect of mutation of F100A on structure-dynamics of the catalytic subunit, we prepared three systems: The wt C-subunit (ATP/Mg$^{2+}$) where Mn$^{2+}$ ions were replaced with Mg$^{2+}$ and removed the PKI to form a binary complex. In addition, we kept the PKI along with the ATP to form a ternary complex (ATP/Mg$^{2+}$/PKI); the F100A mutant structure in both the binary and ternary complex by replacing the Phe100 to Ala, and allowing the side chains to be added by the program LEaP module in AMBER, and the Apo system, which was prepared by taking the coordinates from the PDB: 1J3H (*Akamine et al., 2003*). Amber ff14SB (*Maier et al., 2015*) force field was used to describe proteins. Titratable residues were protonated at pH 7.0 based on PROPKA3.0 (*Olsson et al., 2011*; *Søndergaard et al., 2011*). Parameters for ATP and Mg$^{2+}$ were obtained from the Bryce Group AMBER parameters database (*Meagher et al., 2003*; *Allnér et al., 2012*). The phosphorylated serine and threonine residues were described using the phosaa10 force field (*Homeyer et al., 2006*). The hydrogens and counter ions were added, and the system was solvated in the octahedron periodic box using the TIP3P (*Price and Brooks, 2004*) water model and 150 mM NaCl with a 10 Å buffer in AMBER tools. Systems undergo minimization, heating, and equilibration steps using the AMBER16. Initially, systems were minimized by 1000 steps of hydrogen-only minimization, 2000 steps of protein and ligand minimization, 2000 steps of side chain minimization, 2000 steps of backbone minimization, and followed by removing all restraints for 5000 steps of all-atom minimization. Systems were

gradually heated from 0° to 300° K over 50 ps with 2-fs time-steps and 10.0 kcal mol$^{-1}$Å$^{-2}$ position restraints on the protein. The temperature was maintained by the Langevin thermostat (*Loncharich et al., 1992*; *Pastor et al., 1988*) while the pressure was maintained using the Barendsen barostat (*Berendsen et al., 1984*). Constant pressure equilibration with a 10 Å non-bonded cutoff with Particle Mesh Ewald (PME) (*Darden et al., 1993*) was performed with 1000 ps of unrestrained equilibration. Production simulations were performed on Graphic Processing Unit-enabled AMBER16 as above in triplicate for 200 ns each for an aggregate of 600 ns. Overall, 60,000 snapshots were generated for each system and saved for analysis.

## LSP-alignment-based protein residue networks

LSP-based PRNs were built as described earlier (*Kornev et al., 2022*). Five 10 ns intervals were taken from a 200 ns trajectory at specific intervals: 0–10 ns, 50–60 ns, 90–100 ns, 130–140 ns, and 170–180 ns. From each interval, 100 structures were extracted with a step of 0.1 ns. The LSP-alignment between each set of 100 structures was performed in an all-to-all manner. The resulting adjacency matrices were averaged for each set. Degree centrality and betweenness centrality were calculated for five average matrices. Finally, five values for the centralities were averaged, and the standard error of the mean was calculated.

## PCA and free energy landscape (FEL)

PCA is a broadly used method to extract the slow and functional motions of biomolecules (*Balsera et al., 1996*; *Bahar et al., 1998*; *Maisuradze et al., 2009*). First, the covariance matrix, C, was calculated based on the fluctuations of the Ca atom of each residue. The elements $C_{ij}$ in the matrix were obtained from the fluctuation of amino acids and diagonalized as given in *Equation 1*

$$C_{ij} = < \left( x_i - <x_i> \right) \left( x_j - <x_j> \right) >$$ (1)

where $x_i$ and $x_j$ are the i$^{th}$ or j$^{th}$ atom coordinates and $<x_i>$ and $<x_j>$ represent the mean average coordinate of the i$^{th}$ or j$^{th}$ atom, respectively. The principal components (PCs) will be obtained by diagonalizing the covariance matrix C. The corresponding eigenvalues and eigenvectors were calculated.

The FEL can be drawn based on PCA as a reaction coordinate using the *Equation 2* (*Frauenfelder et al., 1991*; *Maisuradze et al., 2010*)

$$G_i = -k_B T \ln \frac{N_i}{N_m}$$ (2)

where $k_B$ represents the Boltzmann constant and T represents the absolute temperature. $N_i$ and $N_m$ are the i$^{th}$ bin populations and the maximum populated bin, respectively.

## Acknowledgements

This work was supported by the National Institutes of Health, GM100310 and HL144130 to G.V, and GM130389 to S.S.T.

## Additional information

### Funding

| Funder | Grant reference number | Author |
| --- | --- | --- |
| NIH Office of the Director | GM130389 | Susan S Taylor |
| NIH Office of the Director | GM100310 and HL144130 | Gianluigi Veglia |

The funders had no role in study design, data collection and interpretation, or the decision to submit the work for publication.

## Author contributions
Jian Wu, Conceptualization, Formal analysis, Writing – original draft, Writing – review and editing; Nisha A Jonniya, Formal analysis, Writing – original draft, Writing – review and editing; Sophia P Hirakis, Writing – review and editing; Cristina Olivieri, Formal analysis, Writing – review and editing; Gianluigi Veglia, Alexandr P Kornev, Conceptualization, Formal analysis, Writing – review and editing; Susan S Taylor, Conceptualization, Formal analysis, Supervision, Funding acquisition, Writing – original draft, Writing – review and editing

## Author ORCIDs
Jian Wu ⓘ https://orcid.org/0000-0002-8031-9462
Gianluigi Veglia ⓘ https://orcid.org/0000-0002-2795-6964
Susan S Taylor ⓘ https://orcid.org/0000-0002-7702-6108

Reviewer #1 (Public Review): https://doi.org/10.7554/eLife.91980.3.sa1
Author response https://doi.org/10.7554/eLife.91980.3.sa2

# Additional files

## Supplementary files
• MDAR checklist

## Data availability
All data generated or analysed during this study are included in the manuscript and supporting files. Data are available at this link: https://doi.org/10.5061/dryad.c59zw3rjm.

The following dataset was generated:

| Author(s) | Year | Dataset title | Dataset URL | Database and Identifier |
|---|---|---|---|---|
| Taylor S, Jonniya N, Kornev A | 2024 | Role of the α C-β4 Loop in Protein Kinase Structure and Dynamics | https://doi.org/10.5061/dryad.c59zw3rjm | Dryad Digital Repository, 10.5061/dryad.c59zw3rjm |

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
