## [Editor Report · eLife assessment]

This **valuable** study draws attention to the importance of a previously overlooked structural motif in kinase regulation. While the data presented are intriguing and mostly **solid**, further analysis and additional experiments will be needed in the future to support the authors' hypothesis. The work will be of interest to protein biochemists and enzymologists with an interest in kinases and allostery.

---

## [Referee Report · Reviewer #1 (Public Review)]

In this work Wu, J., et al., highlight the importance of a previously overlooked region on kinases: the αC-β4 loop. Using PKA as a model system, the authors extensively describe the conserved regulatory elements within a kinase and how the αC-β4 loop region integrates with these important regulatory elements. Previous biochemical work on a mutation within the αC-β4 loop region, F100A showed that this region is important for the synergistic high affinity binding of ATP and the pseudo substrate inhibitor PKI. In the current manuscript, the authors assess the importance of the αC-β4 loop region using computational methods such as Local Spatial Pattern Alignment (LSP) and MD simulations. LSP analysis of the F100A mutant showed decreased values for degree centrality and betweenness centrality for several key regulatory elements within the kinase which suggests a loss in stability/connectivity in the mutant protein as compared to the WT. Additionally, based on MD simulation data, the side chain of K105, another residue within the αC-β4 loop region had altered dynamics in the F100A mutant as compared to the WT protein. While these changes in the αC-β4 loop region seem to be consistent with the previous biochemical data, the manuscript can be strengthened with additional experiments.

Comments on the revised version:

Additional experiments (both computational and experimental) assessing the role of the αC-β4 loop region (especially residues such as K105) are needed to bolster their hypothesis. My initial assessment therefore remains unchanged. While this manuscript falls short of expectations when it comes to experimental findings, it is an excellent review on the structural elements of kinases and how the newly identified αC-β4 loop region integrates with these important regions. Perhaps the experimental section (LSP analysis and MD simulation data) could be removed and this manuscript could be converted into a Review Article?

---

## [Author Response]

The following is the authors’ response to the current reviews.

We agree with Reviewer #1 that it is not typical to include primary data in a review, but this seems to be a very unusual situation and it is not unprecedented. We seriously believe that it will significantly dilute the impact of the message if we were to separate this into two papers. We intended initially to do a comprehensive review of the αC-β4 motif as we think it is an extremely important element of secondary structure that has been rather overlooked in the protein kinase field. It is the site where the nucleotide and peptide/protein binding sites converge in the C:PKI complex and also in the RIα holoenzyme, which is also a pseudo-substrate inhibitor. This stable element is highly conserved in all protein kinases, and we think it is an extremely important allosteric site where the kinases differ. Thus, it is highly relevant for this set of Elife papers on kinase allostery. In parallel, we have developed the Local Spatial Pattern (LSP) alignment method for identifying Protein Residue Networks (PRNs) into a robust tool. When the Veglia team, our long-time collaborators, did their NMR analysis of the F100A mutant, which is in the αC-β4 loop, we thus decided to do the LSP analysis. The LSP results were so interesting and striking that we decided immediately to explore the motif further and to specifically compare the various crystal structures that we had solved in the past to see if indeed we had missed some changes. In addition to looking at the backbone, we decided to also look at the side chains and to compare the structures with the simulations. The results proved to be extremely informative and defined a multi-pronged approach that could be used to screen any disease mutation or alternatively as an Ala scan for any residue in any protein. I consider this to be one of the most important papers that I have published in many years. It describes a process for exploring the potential dynamic impact of any disease mutation or any point mutation. We emphasize repeatedly that the hypotheses generated from the computational screen will need to be validated experimentally, but our LSP analysis is a rapid and relatively inexpensive way to screen a set of mutations and predict which will have the greatest impact on dynamics. It is an especially powerful and robust way to identify allosteric sites as the LSP approach maps global changes of a single mutation across the entire protein. These mutants would then be prioritized for experimental follow-up. We are indeed now implementing this more comprehensive strategy in two ways. We are specifically exploring three disease mutations in the αC-β4 loop and, in parallel, are also doing a computational Ala scan of the entire loop (L95-L106); however, this is part of a separate and more comprehensive study that will take much longer. It will be the "Proof-of-Principle” of the hypotheses that we propose in our Elife paper. In addition to the LSP method, the MD simulations provide new and complementary insights into side chain dynamics in contrast to the static crystal structures. We will also begin to compare the αC-β4 loop in other kinases, specifically PKCβ2 and LRRK2, but once again this is part of a separate study and is clearly beyond the scope of this Elife paper. This focus on the αC-β4 loop is an excellent strategy that can be applied to any protein kinase. The LSP approach, however, can obviously also be applied to any protein or any motif, so it is potentially very powerful tool. We think that the impact and potential importance of this paper will be lost if it is split into two papers.

I went back to look at a recent review that we did for the Biochemical Journal on the PKA Cβ isoform, and there we also included some new primary data in the review. It was never questioned. We believe that our manuscript is so perfectly appropriate for this Elife series that is focus on allostery in kinases, and having our paper back-to-back with the Veglia NMR paper is especially important and relevant. We thus ask you will seriously consider keeping this as a single paper as part of this series on allostery.

The following is the authors’ response to the original reviews.

**Public Reviews:**

**Reviewer #1 (Public Review):**
In this work Wu, J., et al., highlight the importance of a previously overlooked region on kinases: the αC-β4 loop. Using PKA as a model system, the authors extensively describe the conserved regulatory elements within a kinase and how the αC-β4 loop region integrates with these important regulatory elements. Previous biochemical work on a mutation within the αC-β4 loop region, F100A showed that this region is important for the synergistic high affinity binding of ATP and the pseudo substrate inhibitor PKI. In the current manuscript, the authors assess the importance of the αC-β4 loop region using computational methods such as Local Spatial Pattern Alignment (LSP) and MD simulations. LSP analysis of the F100A mutant showed decreased values for degree centrality and betweenness centrality for several key regulatory elements within the kinase which suggests a loss in stability/connectivity in the mutant protein as compared to the WT. Additionally, based on MD simulation data, the side chain of K105, another residue within the αC-β4 loop region had altered dynamics in the F100A mutant as compared to the WT protein. While these changes in the αC-β4 loop region seem to be consistent with the previous biochemical data, the results are preliminary and the manuscript can be strengthened (as the authors themselves acknowledge) with additional experiments. Specific comments/concerns are listed below.1. MD simulations were carried out using a binary complex of the catalytic subunit of PKA and ATP/Mg and not the ternary complex of PKA, ATP/Mg and PKI. MD simulations carried out using the ternary complex instead of the binary complex would be more informative, especially on the role of the αC-β3 loop region in the synergistic binding of ATP/Mg and PKI.

Response 1. Thank you for your suggestion. We have included the data for the MD simulations of the ternary complex in the revised manuscript. This includes a new figure and was indeed informative (Figure 11). Text describing this simulation is also added on pages 15-17. All the changes in the revised manuscript are highlighted in red.

1. The LSP analysis shows a decrease in degree centrality for the αC-β4 loop region in the F100A mutant compared to the WT protein which suggests a gain in stability in this region for the F100A mutant (Fig. 8A). These results seem to be contradictory to the MD simulation data which shows the side chain dynamics of K105 destabilizes the αC-β4 loop region in the F100A mutant (Fig. 10B). It would be helpful if the authors could clarify this apparent discrepancy.

Response 2. In Figure 8A, the negative values of degree centrality for the αC-β4 loop region show that the value of DC is less in the WT compared to the mutant, suggesting that those regions are more stable in the mutant. This says that the mutation in the αC-β4 loop region both rigidifies the motif and alters the communication signaling networks between the two lobes.

The betweenness centrality plots (Figure 8B) also show how the connectivity between the two lobes is altered upon mutation. In the mutant the major connectors become V104 and I150 in the C-lobe, whereas connectivity was primarily governed by K72 (N-lobe) and D184 (C-lobe) in the wt C-subunit. Overall, the mutation causes rigidification of the αC-β4 loop and this leads to loss of allosteric communication between the two lobes.

The MD simulation results as shown in Figure 10B are not contradictory. This figure shows the overall dynamic profile of the protein, based on principal component analysis (PCA) using the parameter of the residual flexibility. It does not reflect a particular motif's stability or flexibility. Instead it shows that overall the protein upon mutation becomes more dynamic and can sample different conformational states, while, in contrast, the WT protein preferred a single global state of conformation. However, the LSP results showed that, compared to the other parts, the αC-β4 loop, especially V104 at the tip, becomes more stable following mutation, and this has an impact on the allosteric communication between the two lobes.We have added this information into the revised manuscript on page 14, also highlighted in red.

1. The foundation for the experiments carried out in this paper are based on previous NMR and computational data for the F100A mutant. However, the specific results and conclusions from these previous experiments are not clearly described.

Response 3. The NMR paper has been already accepted by eLIFE and here we are attaching the bioRxiv paper link, “https://www.biorxiv.org/content/10.1101/2023.09.12.557419v1.”

**Reviewer #1 (Recommendations For The Authors):**
In this work Wu, J., et al., draw attention to the αC-β4 loop, a previously neglected region within kinases. A comprehensive review on the important regulatory elements within the kinase along with how the αC-β4 loop (and the αE helix) integrates with these different regulatory elements is presented well. As the authors themselves acknowledge, the data presented here while promising is preliminary. Additional biochemical, NMR and computational experiments need to be carried out to assess the importance of F100, K105 and other residues in this region.1. The authors indicate that previous computational studies predict a flip in the αC-β4 loop in the apo state. It would be helpful to have a figure showing the predicted flip as well as an explanation for the significance of this predicted flip.

Response 1. The NMR paper has been already accepted by eLIFE and here we are attaching the bioRxiv paper link, “https://www.biorxiv.org/content/10.1101/2023.09.12.557419v1.” The Figures 3 and 6 in that paper described the predicted flip in the αC-β4 loop in the apo state. We did not see a flip in any of our crystal structures, and the LSP analysis which is based on 200 ns simulations is not sufficient to see this major conformational change.

1. The authors cite previous NMR and biochemical experiments (reference 62), work that has just been submitted to eLife. Access to this work was difficult as this manuscript could not be found on the eLife website.

Response 2. The NMR paper has been already accepted by eLIFE and here we are attaching the bioRxiv paper link, “https://www.biorxiv.org/content/10.1101/2023.09.12.557419v1.”